# Human Bocavirus in Childhood: A True Respiratory Pathogen or a “Passenger” Virus? A Comprehensive Review

**DOI:** 10.3390/microorganisms11051243

**Published:** 2023-05-09

**Authors:** Sandra Trapani, Alice Caporizzi, Silvia Ricci, Giuseppe Indolfi

**Affiliations:** 1Department of Health Sciences, University of Florence, Viale Pieraccini, 24, 50139 Florence, Italy; 2Pediatric Unit, Meyer Children’s Hospital IRCCS, Viale Pieraccini, 24, 50139 Florence, Italy; 3Division of Immunology, Meyer Children’s Hospital IRCCS, Viale Pieraccini, 24, 50139 Florence, Italy; 4NEUROFARBA Department, University of Florence, Viale Pieraccini, 24, 50139 Florence, Italy

**Keywords:** human bocavirus, immune response, respiratory tract infection, co-infection, children

## Abstract

Recently, human bocavirus (HBoV) has appeared as an emerging pathogen, with an increasing number of cases reported worldwide. HBoV is mainly associated with upper and lower respiratory tract infections in adults and children. However, its role as a respiratory pathogen is still not fully understood. It has been reported both as a co-infectious agent (predominantly with respiratory syncytial virus, rhinovirus, parainfluenza viruses, and adenovirus), and as an isolated viral pathogen during respiratory tract infections. It has also been found in asymptomatic subjects. The authors review the available literature on the epidemiology of HBoV, the underlying risk factors associated with infection, the virus’s transmission, and its pathogenicity as a single pathogen and in co-infections, as well as the current hypothesis about the host’s immune response. An update on different HBoV detection methods is provided, including the use of quantitative single or multiplex molecular methods (screening panels) on nasopharyngeal swabs or respiratory secretions, tissue biopsies, serum tests, and metagenomic next-generations sequencing in serum and respiratory secretions. The clinical features of infection, mainly regarding the respiratory tract but also, though rarely, the gastrointestinal one, are extensively described. Furthermore, a specific focus is dedicated to severe HBoV infections requiring hospitalization, oxygen therapy, and/or intensive care in the pediatric age; rare fatal cases have also been reported. Data on tissue viral persistence, reactivation, and reinfection are evaluated. A comparison of the clinical characteristics of single infection and viral or bacterial co-infections with high or low HBoV rates is carried out to establish the real burden of HBoV disease in the pediatric population.

## 1. Introduction

Human bocavirus (HBoV, lately denoted HBoV1) is a parvovirus that was isolated ten years ago, mainly affecting the lower respiratory and gastrointestinal tracts in childhood all over the world. It is a small, icosahedral, linear, non-enveloped, single-stranded DNA virus measuring between 18 and 26 nm [1]. It was first identified by Allander in 2005 in 17 respiratory samples from children suffering from acute respiratory tract infections (RTIs) of suspected viral origin in Sweden [2]. Since 2009, other strains of the virus have been detected: HBoV2, HBoV3, and HBoV4 [3,4,5].

HBoV1 has been found primarily in samples from the respiratory tract, responsible for mild to severe upper and lower RTIs; in contrast, the variants (HBoV 2–4) have been identified in gastrointestinal tract samples and seem to be involved in the pathogenesis of gastroenteritis [6]. Their transmission is most likely to occur via the respiratory and fecal–oral routes [7]. Although HBoV 2–4 variants have also been found in respiratory tract samples at lower frequencies, their role in the pathogenesis of respiratory infections is unclear. Moreover, HBoVs have been detected in urine [8], saliva [9], blood [10,11], tonsils [12,13], cerebrospinal fluid [14], and in environmental samples such as river water [15], sewage [16,17], and shellfish [18] with uncertain clinical significance.

The name *Bocavirus* derives from a combination of the words bovine parvovirus and canine minute virus, which have similar genetic and amino-acid structures [2,19].

Bocavirus usually occurs in infants and children aged between 6 and 24 months, but sometimes it has been found in children older than 5 years and adults [20].

HBoV can be detected alone or, more commonly, with other viruses causing respiratory or gastrointestinal infections, such as rhinoviruses, adenoviruses, noroviruses, and rotaviruses [1,4]; viral co-infection with the respiratory syncytial virus (RSV) is frequently found [1,21]. Furthermore, HBoV can be associated with bacterial co-infections, often with a severe course. HBoV1 has also been detected in asymptomatic children. The high rates of co-infections and the presence of HBoV1 in asymptomatic individuals could explain the high prevalence of the virus in the pediatric population. These data have long raised the question of its involvement in the pathogenesis of respiratory infections [22,23,24]. Many reports, however, highlight the HBoV involvement in life-threatening respiratory illness in children, although a definitive pathogenic role has not been demonstrated [25,26,27].

The pathogenicity of HBoV remains questionable because often this agent seemed to be a harmless passenger rather than a true pathogen [28,29]. However, in the most recent studies, there is good evidence supporting the hypothesis that HBoV is a genuine pathogenic agent, also when it is the sole infectious agent [30,31].

Given the potential for serious and even fatal consequences from such an infection, promptly diagnosing an acute HBoV1 infection is essential for pediatricians.

## 2. Epidemiology

Several clinical studies have identified HBoV1 as one of the most detected respiratory viruses in young children with RTIs and, as with other respiratory viruses, it is widespread in both low- and high-income countries [1,2,3]. Most studies showed a higher prevalence in males [1,4,5,6,7,8,9], others did not find any sex predominance [10], and a few showed a slightly higher prevalence among females [3,11,12].

Most literature reported HBoV infection as prevalent in children between 6 months and 2 years with the highest detection rate during the second year [4,6,10,12,13,14], while it is rare (<5%) during the first 6 months of life [15,16]. Only in the study by Wang et al. were children > 5 years the most affected, followed by children < 2 years and infants [1,15,16].

Infection can occur at any time of the year, with the highest incidence rate in winter and spring [3,17,18,19,20]. In a recent report from Sweden, Oldhoff et al. showed a peak between December and March [5], as with the Turkish cohort reported by Ademhan Tural et al., which had the highest incidence (71.3%) between November and February [21], and Alkhalf et al. who found that most cases occurred during the first three months of the year, and fewer cases were seen during summer [10]. In contrast, in China, the incidence of HBoV infection peaked in summer and autumn [4,22]. Seasonal distribution in subtropical regions appears to differ from temperate zones with two prevalence peaks, one between June and September and one between November and December [23].

The prevalence of HBoV depends on climatic and geographical factors, the type of sample, and the detection method used.

The current worldwide prevalence of HBoV has been estimated to be 6.3% for respiratory infections and 5.9% for gastrointestinal ones [24]. However, numerous studies from different countries revealed a wide range in its prevalence. In China, two studies by De et al. and Tang et al. on children hospitalized for lower RTIs found a detection rate of 6.88% [25] and 6.62% [4], respectively, with a significantly higher number of infected children aged between 1 to 3 years than in other age groups [4]. In a smaller Chinese cohort of 199 cases with RTIs [1], HBoV was the most common virus, found in 28.1% of subjects, significantly higher than the 10% rate reported by Ji et al. who tested 878 children with community-acquired pneumoniae (CAP) [6].

A recent systematic review on the epidemiology of HBoV in the Middle East and North Africa showed a mean value of prevalence in children of 3.94%, with the lowest in Iran (0%) and the highest in Egypt (56.8%) [26]. In other countries, this rate greatly varied: 16.5% was reported by Symekher et al. among Kenyan children [7], 8.1% in children with RTIs in Nigeria [3], 7.2% in Vietnam [27], 7% in Saudi Arabia [12], and 1.9% in Kuwait [28].

In Europe, a recent meta-analysis by Polo et al., including 35 studies on 32,656 subjects from 16 countries, showed that HBoV prevalence varied from 2.0% to 45.69% with a pooled estimate rate of 9.57% [29]. More recently, a retrospective analysis through *PCR* screening on an English cohort of children with RTIs identified a 9% prevalence of HBoV1 [20]; in Croatia, a recent study on 957 children hospitalized with RTIs revealed a prevalence of 7.6% [30], and, in Norway, the prevalence reached 12% in a study by Christensen et al. [31]. The lowest rates were found by Pinana in Spain (3%) [8], and Kantola et al. in Finland (2%) [32].

Only limited data are available on the prevalence of HBoV in asymptomatic individuals since most studies have focused on children with different clinical symptoms of infectious diseases. However, recent publications described the detection of viral genomes in about 5% of respiratory samples obtained from asymptomatic children [33,34]. Nevertheless, asymptomatic subjects had significantly lower viral loads than symptomatic ones as demonstrated by Ghietto et al. [35].

The exact prevalence of acute HBoV infections can only be determined using serodiagnosis and combining at least the *PCR* and serological results. Maternal antibodies have been detected in 90% of infants younger than 3 months old, after which seropositivity decreases and reaches a low level by the age of 6–12 months [36]. Then, HBoV seroprevalence increases until the age of 6 years, when seroprevalence is already around 80% for HBoV1, 50% for HBoV2, and 10% for HBoV3 [37].

## 3. Viral Structure, Transmission, and Pathogenesis

Bocaviruses are small non-enveloped viruses, with linear ssDNA genome. Because the genome has a bilateral hairpin structure at both ends, the full length of its genome cannot be accurately determined, although it has been estimated to be around 4.7–5.7 kb (5543 nucleotides) [38].

There are three open-reading frames (ORF1, ORF2, and ORF3): ORF1 (on the left half) encodes non-structural proteins NS1–4; ORF2 is a smaller middle reading frame which encodes a unique nuclear protein, the nuclear phosphoprotein NP1, essential in viral DNA replication and mRNA processing [39]; and ORF3 (on the right half) encodes structural capsid proteins VP1, VP2, and VP3, with VP3 being the major capsid protein. On one hand, NS1 and NP1, which seem to be relatively conserved, are essential for viral DNA replication and are employed as a target for HBoV detection [38,40]; on the other hand, VP1 and VP2 have greater mutational diversity and are commonly used for the phylogenetic analysis of HBoV [17]. VP1 is essential for infectivity and is facilitated by the release of the virus from endocytic compartments to the nucleus of the host cell [41]. Based on the nucleotide divergence of the VP1 capsid region, the virus has been classified in the four above-mentioned genotypes: HBoV-1, -2, -3, and -4. The viral non-enveloped capsid surface carries determinants of the host and is involved in many processes, including host tropism, cell recognition, pathogenicity, intracellular trafficking, genome packaging, assembly, and the immune response [42,43].

The virus can enter the host via the respiratory tract, through the bloodstream or via the fecal–oral route by direct ingestion [44]. Vertical transmission of human bocaviruses from mother to fetus has not been described [2].

The pathogenesis of human bocaviruses has been poorly studied [45]. The HBoV1 virions can persistently infect human airway epithelial cells from both the apical and basolateral surfaces. A consequence of infection can be the induction of airway epithelial damage [46]; the viral replication inside cells, although without high replicative activity, causes a loss of cellular integrity, loss of cilia, rupture of cell junctions, epithelial cells hypertrophy, and persistent infection, as well as cell death induced by caspase-1 activation. These findings suggest that the pathogenesis of HBoV is related to the disruption of the epithelial barrier function. HBoV1 infection causes the death of airway epithelial cells, resulting in airway injury and inflammation [38,46].

Besides the acute infection, HBoV is known to persist for a long time in lymphatic tissue and tissues afflicted with chronic sinusitis [47,48].

## 4. Immune Response

Current knowledge of the adaptive immune response, including both T- and B-cell-based cellular immunity evoked in response to HBoV infection, is inconclusive. HBoV viral capsid proteins (VP1 and VP2) have been described as immunodominant targets of the cellular and humoral immune response [24,33]. High seroprevalence of HBoV1-VP1-specific IgG has been observed in young children, while HBoV-VP2-specific T-cell responses, mostly CD4-mediated and enhanced IFN-γ, have been observed in adults.

During the acute phase of the HBoV infection, the T-cell response is mediated mostly by CD4+ T cells rather than CD8+ T cells. The NK ratio has been found to be low, whereas the CD19+ CD23+ ratio, which is a marker of B-cell activation, is high, as expected for viral pathogenesis [4]. A comparison of the Th1/Th2-type cytokine profiles shows high concentrations of IFN-gamma, IL-2, and IL-4 in children with HBoV-related bronchiolitis [49] and in vitro stimulation of CD4 T cells with HboV1 virus-like particles (VLP) which increases the secretion of IFN-γ (Th1) and IL-10 and IL-13 (Th2). The induction of Th1 cytokines contributes to virus elimination whereas Th2 proinflammatory ones may contribute to asthma exacerbations associated with HBoV infection [18]. Further study on this issue demonstrated that several cytokines associated with lung fibrosis and tumor development, e.g., EGF, VEGF, TNF-α, TNF-β, and TIMP-1, were upregulated in the HBoV-positive cohort. These data suggest that the development of lung fibrosis might be triggered by HBoV-induced cytokine expression [50].

After the acute phase, HBoV may escape the host’s immune surveillance and establish a silent persistence. The immune-evasion strategies of HBoV involve type I IFN production or a TNF-a response. It has been demonstrated that HBoV nuclear phosphoprotein (NP1) suppressed IFN-β production by targeting the IRF-3 signalling pathway [51], while nonstructural proteins (NS1 and NS1-70 proteins) act as antagonists of the NF-kB pathway, nullifying the response to TNF-a [52].

Recent data published by Ivaska et al. analyzing tonsil tissue samples showed distinctively decreased T-helper17- and T-regulatory-type immune responses in local lymphoid tissue in HBoV-positive patients who underwent tonsillectomy. The authors demonstrated an inverse correlation between intra-tonsillar HboV1-DNA loads and cytokine expressions (INF I and II, IL-28, IL-29, and IL-13) [53]. Together, the immune modulator mechanisms of HBoV manipulate the host’s immune surveillance, allowing persistent virus replication and propagation. A recent study by Xu et al. provides new insights into tonsillar HBoV1 persistence; the authors found HBoV1 persistence only in germinal centers where immune maturation occurs, and the main host cells were B cells and monocytes. In these cells, the virus uptake was significantly enhanced by HBoV1-specific antibodies, mediated by the cellular IgG receptor, leading to viral mRNA synthesis [54]. The HBoV immune response is schematized in Figure 1.

## 5. Risk Factors

Pre-existing comorbidities which may represent a risk factor for a severe course of the disease [2], such as chronic lung diseases, congenital heart diseases, neuromuscular disorders, cancer, or immunologic conditions, are commonly found in HBoV-infected children with rates that range from 14% [5,53] to 77% [21].

Furthermore, prematurity was reported in 11.6% of cases by Campelo et al., who, additionally, found that passive smoking was the principal risk factor in his cohort [55]. HBoV1 was also associated with increasing age, winter, and childcare attendance in the Observational Research in Childhood Infectious Diseases study on an Australian cohort of children younger than 2 years [15]. The most recent meta-analysis indicated that being <5 years old is a risk factor for HBoV infection [29]. Other risk factors for HBV1-induced respiratory illness in infants are maternal smoking, being born in winter, and a family history of asthma [56].

## 6. Clinical Features

### 6.1. Respiratory Features

HBoV infection is described as affecting the lower respiratory tract predominantly [3,6,10,57,58,59]. HBoV infections have been detected in children with acute diseases of the upper and lower respiratory tract, often combined with interstitial lung infiltrate and abnormal radiologic findings. Respiratory manifestations, particularly cough, stuffy and runny nose, pharyngeal hyperemia, wheezing, tachypnea, shortness of breath, hypoxia, cyanosis, and chest retractions are the most reported signs or symptoms [1,6,10,59]. Clinical diagnoses include pneumonia, bronchiolitis, bronchitis, rhinitis, tonsillitis, laryngeal croup, and, more rarely, conjunctivitis. In a systematic review and meta-analysis of common respiratory viruses in infants with bronchiolitis, HBoV is reported as the third most common virus [60]. Another rare pulmonary disease, named “plastic bronchitis” has occasionally been associated with HBoV infection [61,62]. HBoV1 can also exacerbate chronic pulmonary diseases such as asthma, chronic obstructive pulmonary disease, and cystic fibrosis [2]. In one Spanish cohort, early severe HBoV infection was associated with the development of asthma in children aged 5–7 years [63].

### 6.2. Gastrointestinal Features

The variants (HBoV 2–4) are currently associated with gastrointestinal symptoms, the most frequent being the loss of appetite and vomiting, followed by diarrhea and nausea [10]. Co-infections with intestinal pathogens (e.g., human rotavirus, noroviruses, and enteropathogenic strains of *Escherichia coli* or *Salmonella* sp.) have been frequently reported, with up to 77.6% of HBoV-positive children. However, the nature of the relationship between HBoV and gastroenteritis remains unclear [64].

### 6.3. Systemic Manifestations

Fever has been frequently reported, varying from 68% by Tang et al. [4], 65% by Ji et al. [6], 52% by Zhang et al. [57], 46% by Petrarca et al. [19], to 41% by Wang et al. [1]. Other systemic manifestations included fatigue, lymphoadenomegaly, splenomegaly, and skin rash [10].

### 6.4. Complications

The most reported complications related to bocavirus infection were increasing respiratory distress leading to acute respiratory failure, followed by dehydration [10]. In 878 children with CAP, Ji et al. observed that those infected with HBoV were more likely to progress to severe pneumonia [6]. Pneumomediastinum and bilateral pneumothorax are other rare and life-threatening conditions developing in bocavirus-infected children [10]. Besides respiratory failure, HBoV can be also associated with acute heart failure at a significant rate (10%), as reported by Zhang [59]. In a few cases, hepatitis and myocarditis were also reported [65,66].

Liao et al. reviewed 12 children with HBoV1-related severe respiratory infection admitted to the ICU, including a case described by himself [67,68,69,70,71,72,73,74,75,76]. The diagnoses were respiratory failure (five cases), ARDS (five cases), bronchiolitis, atelectasis, and status asthma, suggesting that severe HBoV infection often leads to respiratory failure and ARDS [72,77]. Of note, five of them (45%) died [77]. Other fatal cases have been described, such as a child with a mono-infection and pre-existing asthma in a Swedish cohort [5] and a 13-month-old child with myocarditis associated with HBoV2 [66].

Unfrequently, the central nervous system is affected in children with HBoV infections with symptoms including seizures, encephalitis, and encephalopathy [78,79,80,81]. Guillain–Barré syndrome has been reported in a case of coronavirus co-infection [5], and a case of acute encephalopathy with biphasic seizures and late restricted diffusion associated with HBoV infection has been recently described in a female infant [82].

The severity of the illness could be remarkable. In the study by Akturk et al., 33.3% required PICU admission, and 20% were mechanically ventilated [81]. Several papers indicated that a high HBoV viral load (indicated by lower Ct values) is associated with severe disease [83] and longer hospitalization [46]. Jiang et al. reported that respiratory infections where only HBoV was detected had a higher viral load, and were associated with a greater severity of the disease [83]. This result confirmed the previous conclusions by Zhou et al. that HBoV1 at a high viral load is not frequently found with other viruses and that a single detection with a high viral load correlates with severe respiratory disease [84].

In several studies, the median hospitalization length (LOS) was seven days [1,4]. Alkhalf et al. reported the shortest median LOS of 5.31 days, [10], whereas Tural et al. reported the longest one with a median LOS of 13 days, even longer (15 days) for children with a single HBoV infection [21].

## 7. Co-Infections

Co-infection of HBoV with other respiratory viruses or bacteria has been reported at different rates. A high rate of co-detection is a well-recognized characteristic of bocavirus infection, and it is probably the result of prolonged shedding of HBoV1 in the nasopharynx, for weeks or even months [16,85].

### 7.1. Viral Co-Infections

The mixed viral infection rate is relatively high in most studies on NP swab samples, with rates of 38.3% in Australia [15], 47% in Denmark [86], 51.6% in China [4], 54% in Panama [87], around 75% in Italy [88], Spain [8,75], UK [20], and Norway [2], up to 80% in Saudi Arabia [10], 82.2% in Croatia [30], and 84% in Gabon [89]. Guido et al. estimated the rate of HBoV co-infections with other viruses to be 52.4% (respiratory infections) and 46.7% (gastrointestinal infections) [24]. In the largest cohort analyzed by Alkhalf et al., at least 80% of children hospitalized with bocavirus infection had one or more co-viral infections, most commonly human rhinovirus (HRV) (45%), human adenovirus (ADV) (30%), and RSV (7%) [10]. Similar associations were reported by other authors with varying percentages; Ji et al. confirmed that HRV was the most commonly co-infecting virus (20.5%), followed by RSV (17%) [29]. The systematic review of Kenmoe et al. showed that, in children with bronchiolitis, HBoV is mostly associated with HRV and RSV [58], in line with the results of Petrarca et al. [19]. RSV was reported as the most frequent HBoV co-infection in the Middle East and North Africa [26]. In the Chinese group reported by Tang et al., only 26.1% of children had mixed common respiratory viral infections, mostly HRV (11.4%), and RSV (6%) [4]. A similarly low rate was described by Wang et al. who confirmed a co-infection rate of 25% with one or two other viruses, with RSV being the most frequent, followed by HRV [1]. The detection of at least one other gastrointestinal virus in HBoV2- and HBoV3-positive fecal samples from children with gastroenteritis has also been reported in 44–100% of cases [2].

### 7.2. Bacterial Co-Infections

In several studies, a bacterial co-infection was found most frequently for *S. pneumoniae*. In 69.6% of the Chinese HBoV-positive cases, a dual or triple bacterial co-infection has been detected, with *S. pneumoniae* and/or *M. catarrhalis* being the most commonly found (with a detection rate of 32.1% and 23.2%, respectively), followed by H. influenzae for single-infection, and *S. pneumoniae* and *S. maltophilia* for mixed-infections (both with a detection rate of 5.4%) [1]. The detection rate of co-infections with *S. pneumoniae* was 29.9% in the Chinese ARIs Child Surveillance Study for the period 2009–2019 [88], and 13.33% in infants presenting with RTIs in Ningbo by Zhang et al. [59]. In the cohort reported by Tang et al., among children with mixed infections, 19.18% had mixed bacterial infections, including *S. pneumoniae* (10.22%), *H. influenzae* (5.9%), and *S. aureus* in 10 children (1.8%) [4]. A different rate (27%) and spectrum of bacteria were found by Ji et al. in his HBoV-positive group, with *Escherichia coli* being the most frequent (29% of co-infection), followed by *Klebsiella pneumoniae* (20%); the authors also noted that the bacterial co-infections were significantly more frequent in infants less than 1-year-old [6]. *Mycoplasma Pneumoniae* infections have been frequently found, for example, at a rate of 14.5% by Tang et al. [4] and 33% by Ji et al. [6] and Zhang et al. [59].

Some authors note that clinical symptoms are more severe in cases with multiple infections than those with mono-infections [90,91]. Eski et al. found that, in Turkey, children with co-infection had higher risks of ICU admission than those with HBoV mono-infection [92]. Similarly, in the study by Pinana et al., children with co-infections required respiratory support for significantly longer than those with single detection [8]. The presence of multiple respiratory tract co-infections was associated with a severe course and ICU admission also in the Chinese cohort reported by Zhang et al. [59].

There have, however, been cases of life-threatening diseases caused by a mono bocavirus infection [5,72,93] and several fatal cases in infants with lower RTIs [71,75,76]. Similarly, Colazo-Salbetti found pneumonia was more frequent in mono-infections than in co-infections [94]; the same results were obtained by Petrarca et al. who found HBoV detected alone in severe cases and all cases of pneumonia [19]. In accordance, Wang et al., in comparing the clinical characteristics of patients infected by HBoV alone or in combination, noted that the first group had higher values of AST and eosinophils (EOS), and a higher rate of pneumonia [1]. These recent data suggest that HBoV could play a true pathogenetic role in patients with severe lower RTIs, in contrast to the previous hypothesis about a synergic role in multiple respiratory infections [95].

## 8. Detection Methods

The diagnosis of HBoV infection has so far mainly been based on the detection of viral genomes present in human respiratory samples, although serum, stool, and urine samples are also used. Techniques include different molecular assays (polymerase chain reactions, PCR) employing numerous sets of primers specific for the viral genes NP1, NS1, and VP1/2 [33]. The most common methods are quantitative PCR (qPCR) and reverse transcription PCR (RT-PCR) measuring HBoV messenger RNA (mRNA). Samples from the upper (NP aspirates, NP swabs, or oropharyngeal swabs), middle (tracheal aspirate), and lower respiratory tract (broncho-alveolar lavage) are examined in patients with respiratory tract infection. Multiplex tandem PCR (MT-PCR) has been recently introduced to analyze samples in which HBoV was detected in combination with other respiratory viruses; a comparison of MT-PCR with qPCR showed that MT-PCR could be a powerful tool for distinguishing the true nature of HBoV detection based on the amount of virus compared to co-detected pathogens [96].

Stool samples may be the specimen of choice for patients with gastrointestinal symptoms. It is well-known that viral shedding can be prolonged, owing to the persistence of HBoV DNA in the nasopharynx for months and even up to a year, which can hamper initial diagnosis [16,85,97,98]. The viral persistence in the respiratory tract limits the usefulness of molecular diagnosis, which is highly sensitive, but of questionable specificity.

The presence of a high HBoV1 DNA load or mRNA in respiratory secretion is often considered a marker of viral activity, and can be used to distinguish acute infection from DNA persistence [99]: high concentrations (from 10^4^ to 10^8^) of HBoV1 DNA copies/mL of NP secretions, expressed as cycle threshold (Ct) values < 25, are suggestive of acute HBoV1 infection [16]. Oldhoff et al. found that low Ct values (<25) were associated with young age (*p* < 0.001) and single detection of HBoV1 [5]. Single detection and low Ct may be associated with more severe respiratory symptoms, but we only have data for a small number of patients and the results of comparative studies are inconclusive [16,31,99].

Serology has a higher specificity but lower sensitivity [100]. It has been estimated that no more than 20–25% of patients positive for HBoV1 DNA result in having an acute infection from serodiagnosis, and thus, serological data alone is likely to underestimate human exposure to these viruses [101]. However, the detection of HBoV1-specific IgM and an increase or seroconversion of IgG provides a higher specificity than qualitative PCR-based assays [102]. For the diagnosis of an acute primary HBoV1 infection, Christensen et al. recommended the presence of at least two of the following factors: high DNA load (>10^6^ copies/mL) on NP secretions by quantitative PCR or HBoV1 mRNA, serum positive IgM, low IgG avidity, and/or >4-fold IgG titre [2]. To overcome the diagnostic difficulties, we suggest a diagnostic criteria proposal for acute primary HBoV infection, as reported in Table 1.

In conclusion, no test alone is sufficient in all instances; for accurate HBoV1 diagnosis in severe infections, molecular assays and serology of paired serum samples should be combined and considered together with other laboratory data, clinical features, and the time of symptom onset [1,25,100,102].

## 9. Laboratory and Radiological Investigations

### 9.1. Biochemical Tests

C-reactive protein (CRP) and white blood cells are generally normal or just slightly elevated during acute HBoV infection. However, in the study by Tang et al., 36.7% of children with an HBoV single-infection showed elevated CRP levels [4]. The plasmatic concentrations of abnormal lactate dehydrogenase (LDH), urea (URE), creatine kinase isoenzyme (CK-MB), and Mg in HBoV-positive patients were significantly altered compared to those in HBoV-negative patients in the study by Wang et al. [1]. Similarly, the values of aspartate aminotransaminase (AST) were significantly higher in HBoV-positive patients with a median value close to the upper range [1]. The same authors had found similar results also in a previous analysis [103]. In keeping with these data, Ji et al. reported significantly higher LDH values in HBoV-positive children [6]. Given that AST, LDH, CK-MB, and URE are biochemical indices of liver and heart dysfunction [104] and kidney injury, it could be hypothesized that HBoV may affect these organs simultaneously or that these findings are attributable to indirect immune responses of various organs to the pathogen [1].

### 9.2. Imaging

Chest radiography frequently shows peribronchial or interstitial infiltrates, hyperinflation, or atelectasis [105]. In one study, interstitial infiltrates were observed in 75% of children hospitalized with lower RTIs associated with serologically confirmed HBoV1 infections [106]. Zhang et al. described abnormal radiography findings in all HBoV-positive infants: in detail, 65% of infants had pneumonia, 34% acute bronchitis or bronchiolitis, 12% focal pulmonary atelectasis, and 9% localized pulmonary emphysema [59]. Akturk et al. reported that 43.3% of his cases had radiologically confirmed pneumoniae [81].

## 10. Management

Yet, there is no clinically approved specific treatment for bocavirus infection, and no comparative studies on antiviral drugs have been carried out. To date, there has been a single case report of treatment in an immunodeficient child who was treated with cidofovir, specific for herpesviruses; in this case, HHV-6 viremia decreased, and the HBoV infection was successfully eliminated [107].

As for many viruses, supportive therapy remains the mainstay of treatment for HBoV. This includes providing oxygen for hypoxia and bronchodilators for patients with wheezing [108], and antipyretics [10]. The most administered medical therapy was bronchodilators (mostly β2-adrenergic agonists) followed by oxygen therapy given in more than half of cases [8,10,21,55]. Oxygen supplementation has been performed either with noninvasive ventilation (nasal cannula, high-flow nasal cannula, non-invasive mechanical ventilation, or face mask), or, more rarely, with invasive ventilation in PICU [8,10,21,77].

Alkhaff et al., in a study on HBoV-infected Saudi Arabian children, noted that those with co-infections seem less likely to require an oxygen supply, and postulated that co-existence with other viruses could result in a better immune response, a less severe disease course, and a less likely need for oxygen [10]. In contrast, in the cohort by Tang et al., only 17.4% of the HBoV single-infected children required oxygen support, only 3.3% were assisted by mechanical ventilation, and 9.6% were admitted to PICU [4]. Similarly, PICU admission was reported at 11% in the small Italian cohort by Petrarca et al. [19], and at 9% among the cases studied by Oldhoff et al., mainly for respiratory complications and rarely for prolonged seizures [5]. A greater rate of cases requiring PICU admission was reported by Ademhan Tural et al. in Turkey (13.6%) [21], and by Zhang et al. (27%) in China [59]. In a small English cohort (29 cases with HBoV mono-infection) reported by Bagasi et al., 38% of patients received oxygen, 31% needed intensive care, and 17% were supported by mechanical ventilation with one patient dying of multi-organ failure and viral pneumonitis [20].

Steroids, specifically prednisolone, proved ineffective in treating children with bocavirus [108]. However, 36.6% of Tural et al.’s cohort [21], 47.6% of the Arabian children [10], and 53% of Campelo’s cohort received steroids [55].

In most cases, even without bacterial co-infection, empirical intravenous antibiotics were administered at high rates ranging from 93% [8,10,21,55] to 38% [20]. When compared to other respiratory viruses in children under 5 years with CAP, HBoV was most frequently (90%) associated with antibiotics treatment [55]. In some patients, the additional diagnosis of bacterial infection such as otitis or pneumonia warranted an antibiotic. Even in the birth cohort of Australian HBoV-positive, non-hospitalized infants, antibiotics were prescribed for 23.5% of those seeking healthcare with only one having otitis [15]. Therefore, a stewardship antimicrobial program should be implemented to minimize antibiotic overuse in such patients.

## 11. Conclusions

Despite the high prevalence of pediatric HBoV infections worldwide, the virus is still not well-recognized by many clinicians. Failure to detect HBoV as a viral pathogen responsible for RTIs can lead to an inconclusive diagnosis, prolonged hospital stays, and unnecessary antibiotic use, ultimately contributing to a burden on local and global health and the economy. Expanding routine testing through viral respiratory panels which include HBoV could reduce this health burden.

An early and accurate diagnosis of the etiological agent would be helpful for the effective treatment and adequate management of children with dyspnea and/or gastrointestinal diseases. Numerous studies confirm that the detection of HBoV DNA can persist for months; therefore, this does not necessarily indicate a recent infection. To certainly diagnose an acute infection of HBoV, therefore, required a combination of HBoV high DNA load or mRNA on NP secretions and one of the following three conditions: serum positive IgM, low specific IgG avidity, or >4-fold IgG titre [2].

It has long been debated whether bocavirus was an autonomous pathogen or acted as a “passenger” since, in a significant percentage of cases, it has been found associated with other viruses and/or bacteria.

However, although most HBoV infections mostly have a benign course, the infected children (also the mono-infected cases) could have a severe disease course at a not-marginal rate, requiring PICU admission and invasive ventilation [20,21,81]. Of note, several children with a fatal outcome have also been described as having a single infection [71,75,76,77]. These findings support the hypothesis that HBoV may be a causative agent of severe RTIs and pneumonia and not a simple passenger.

## 12. Further Directions

Future lines of research should deepen our knowledge of the pathophysiology, replication mechanism, and persistence of HBoV in tissues such as the tonsils and adenoids.

Another area of future investigation will be the development of more rigorous diagnostic methods, including sequence-independent amplification techniques combined with next-generation sequencing platforms that will enable the rapid and simultaneous detection of new viral agents, including HBoV.

Further studies are also needed to clarify the influence of multiple viral infections on the severity of bronchiolitis and the clinical impact of HBoV in bronchiolitis cases. The high prevalence of HBoV in double and potentially life-threatening infections requires a new in-depth investigation to better understand the prognosis. Overall, a better understanding of the natural course of HBoV infection, the search for an experimental model, and the development of a specific treatment regimen to prevent HBoV infections including vaccines are needed to optimize its management.

Finally, an additional analysis should be performed on the changing epidemiology of the common respiratory pathogens, including HBoV, during and after the COVID-19 pandemic when several non-pharmaceutical interventions such as wearing masks, hand hygiene, and social distancing have been taken for infection control.

## Figures and Tables

**Figure 1 microorganisms-11-01243-f001:**
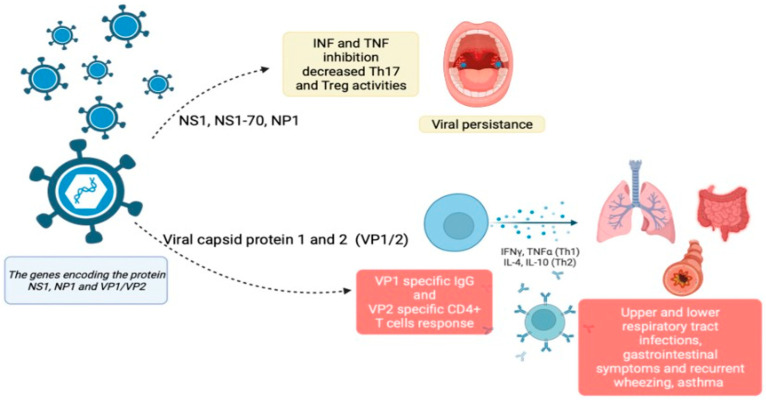
Immune response to HBoV infection: from viral persistence to acute and long consequences. Created with BIORender.com (last updated: 19 March 2023).

**Table 1 microorganisms-11-01243-t001:** Diagnostic criteria proposal for acute HBoV infection.

Nasopharyngeal swab	HBoV high DNA load on secretion
	OR
	HBoV mRNA on secretion
AND one of the following	
Blood	Serum positive IgM
	Low IgG avidity
	>4-fold IgG titre

## Data Availability

No new data were created or analyzed in this study. Data sharing is not applicable to this article.

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
