# Peer review of "Human Bocavirus in Childhood: A True Respiratory Pathogen or a “Passenger” Virus? A Comprehensive Review"

_microorganisms, 2023, doi:10.3390/microorganisms11051243_

Round 1

Reviewer 1 Report

Dear Authors,

The information gathered in the manuscript about HBoV as a serious pathogen or just a "passenger" is very relevant. However, if you ask a question in the title, then the reader expects that the conclusion section will find the answer to this question, which, unfortunately, did not happen. At the end of the manuscript, there is only another statement about the circumstances, due to which the answer to the question cannot be given.

Corrections or additions needed:

1. Line 54: please use the abbreviation for canine minute virus (CnMV). Abbreviation CMV is generally accepted for cytomegalovirus.

2. Line 62: ....The high rates of co-infections and the presence of HBoV1 in asymptomatic individuals could be explained by the persistence and high prevalence of the virus...   

Please explain in more detail how to explain high rates of co-infections and the presence of HBoV1 in asymptomatic individuals with persistence and high prevalence of the virus.

3. Line 227: ....severe HBoV infections were associated with...

There is one infection that can be caused by several pathogenic agents.

4. Line 229: Brackets are missing.

5. Conclusions

The conclusion section needs to be restructured. It definitely needs to be shorter and more to the point. I would recommend shortening the 2 paragraphs from lines 465 to 471 by condensing them into one and showing the main essence.

At this point, it's more of a discussion than a conclusions.

The English language needs minor corrections.

Line 214: HBoV infection is described as affecting predominantly usually presents predominantly lower respiratory tract [3] [10] [6][55] [56] [57].

The sentence is not understood and needs to be restructured.

Author Response

The information gathered in the manuscript about HBoV as a serious pathogen or just a "passenger" is very relevant. However, if you ask a question in the title, then the reader expects that the conclusion section will find the answer to this question, which, unfortunately, did not happen. At the end of the manuscript, there is only another statement about the circumstances, due to which the answer to the question cannot be given.

Your observation is correct; the question in the title requires an answer for the reader. In our opinion, after the extensive review, the conclusion is that HBoV is a serious pathogen and only a “passenger”. Therefore, we modified the conclusion and now we include at the end a statement that confirms our position. 

Corrections or additions needed:

Line 54: please use the abbreviation for canine minute virus (CnMV). 

We remove both the abbreviations (BMV and CMV) as unnecessary; indeed, these words are no more used in the text

Line 62:...The high rates of co-infections and the presence of HBoV1 in asymptomatic individuals could be explained by the persistence and high prevalence of the virus...  

Please explain in more detail how to explain high rates of co-infections and the presence of HBoV1 in asymptomatic individuals with persistence and high prevalence of the virus.

You are right; the above-mentioned statement is incorrect, we modified it as follows: “The high rates of co-infections and the presence of HBoV1 also in asymptomatic individuals could explain the high prevalence of the virus...  

 Line 227: ....severe HBoV infections were associated with...There is one infection that can be caused by several pathogenic agents.

We modified the sentence using the singular “severe HBoV infection was associated with…”

 Line 229: Brackets are missing.

Thank you for the note; the brackets are correctly inserted now

Conclusions

The conclusion section needs to be restructured. It needs to be shorter and more to the point. I would recommend shortening the 2 paragraphs from lines 465 to 471 by condensing them into one and showing the main essence. At this point, it is more a discussion than a conclusion.

The conclusions have been shortened; in particular, the two paragraphs (lines 465- 471) have been condensed.

ENGLISH:

Line 214: HBoV infection is described as affecting predominantly usually presents predominantly lower respiratory tract [3] [10] [6][55] [56] [57]. The sentence is not understood and needs to be restructured. 

The sentence has been restructured as follows: “HBoV infection is described as affecting predominantly the lower respiratory tract”

Reviewer 2 Report

This article describes the epidemiology, clinical and immunological features, and detection methods of human bocavirus infection in an easy-to-understand manner.

Author Response

We thank the reviewer for his/her positive comment

Reviewer 3 Report

A review paper covers very comprehensively  various aspects of Human bocavirus and its role in childhood diseases. The topic is important considering the fact that viruses, including human bocavisus, play an important role in acute infections of the respiratory and digestive tract in children.

Most of the text is clear, although there are chapters that need to be reformatted to make them understandable. Generaly, the sentences are often too long and therefore unclear.

Lines 69-74 – it is not clear what the authors wanted to state.

Lines 81-86 – To much data, but at the end, it remains unclear at what age is the prevalence of HboV infection the highest.

Lines 69-74 – Unclear, should be reworded for better understanding.

Lines 96-98 – The prevalence of HboV does not depend on the mentioned factors - the differences in prevalence stated in various studies are the result of different types of samples and methods of virus detection. The prevalence of HBoV does not depend on the clinical diagnosis.

Lines 214-215 – This is a completely unclear statement.

Clinical features, Co-infections, Laboratory and radiological investigations – the data should be presented so that they can be more easily followed.

Lines 271-277 – Redundant and uninformative

Lines 426-443 – The treatment of respiratory infections caused by HboV does not differ from the treatment of other viral infections. It can be summed up in one sentence as a well-known and accepted fact.

I suggest adding risk factors which can contribute to a more severe clinical picture of HboV infections.

The cited literature is extensive, but it should be supplemented by more recent publications.

Editing  of English language is required.

Author Response

There are chapters that need to be reformatted to make them understandable. Generally, the sentences are often too long and therefore unclear.

Some chapters have been reformatted and divided into subchapters hoping to make them understandable. Many sentences have been shortened.

Lines 69-74 – it is not clear what the authors wanted to state.

We modified the sentences to improve their comprehension.

Lines 81-86 – To much data, but at the end, it remains unclear at what age is the prevalence of HboV infection the highest.

We deleted some data and modify to clarify the message: HBoV infection is prevalent in children between 6 months and 2 years, especially during the second year; however, one author (Wang) reported in his cohort study that children > 5 years were the most affected, followed by those < 2.

Lines 69-74 – Unclear, should be reworded for better understanding.

We reworded as above-mentioned

Lines 96-98 – The prevalence of HboV does not depend on the mentioned factors - the differences in prevalence stated in various studies are the result of different types of samples and virus detection methods. The prevalence of HBoV does not depend on the clinical diagnosis.

We corrected the statement, deleting the clinical diagnosis; the sentence has been rephrased.

Lines 214-215 – This is a completely unclear statement.

The sentence has been restructured as follows: “HBoV infection is described as affecting the lower respiratory tract predominantly”

Clinical features, Co-infections, Laboratory and radiological investigations – the data should be presented so that they can be more easily followed.

These chapters have been reformatted and divided into subchapters hoping to make it easier to follow the data.

Lines 271-277 – Redundant and uninformative.

These sentences have been reduced to a single statement on the median LOS.

Lines 426-443 – The treatment of respiratory infections caused by HboV does not differ from the treatment of other viral infections. It can be summed up in one sentence as a well-known and accepted fact.

The treatment chapter has been reduced, although some issues such as steroids and antibiotics are briefly discussed.

I suggest adding risk factors which can contribute to a more severe clinical picture of HboV infections.

Chapter 5: the risk factors for a severe course of the disease are listed here.

The cited literature is extensive, but it should be supplemented by more recent publications.

A new article has been added to the text and to the references list [94]

Mijac M et al Comparison of MT-PCR with Quantitative PCR for Human Bocavirus in Respiratory Samples with Multiple Respiratory Viruses Detection. Diagnostics (Basel) 2023 Feb 23;13(5):846

Round 2

Reviewer 1 Report

Dear Authors,

Thank you for your corrections, making the manuscript more meaningful and clear.

Author Response

Thank you for your kind response

Reviewer 3 Report

The  paper has been corrected according to the instructions.

I suggest proofreading of the English language by an authorized person

Author Response

Thank you for your appreciation.

The English language has been carefully checked.